# Research on Enhancing the Solubility and Bioavailability of Canagliflozin Using Spray Drying Techniques with a Quality-by-Design Approach

**DOI:** 10.3390/pharmaceutics17101319

**Published:** 2025-10-11

**Authors:** Ji Ho Lee, Seong Uk Choi, Tae Jong Kim, Na Yoon Jeong, Hyun Seo Paeng, Kyeong Soo Kim

**Affiliations:** Department of Pharmaceutical Engineering, Gyeongsang National University, 33 Dongjin-ro, Jinju 52725, Republic of Korea; leejh5117@naver.com (J.H.L.); tlrp26@naver.com (S.U.C.); rlaxowhd476@naver.com (T.J.K.); saera23@gnu.ac.kr (N.Y.J.); paenghyeonseo2202@gmail.com (H.S.P.)

**Keywords:** canagliflozin, solid dispersion, spray drying, quality by design, Box–Behnken design, pharmacokinetic

## Abstract

**Objectives**: The objective of this study was to enhance the solubility and bioavailability of canagliflozin (CFZ) using a spray drying technique with a Quality-by-Design (QbD) approach. **Methods**: The formulation of CFZ-loaded solid dispersions (CFZ-SDs) was optimized using a Box–Behnken design (BBD) with three factors at three levels, resulting in a total of fifteen experiments, including three central point replicates. The design space was determined using the BBD, and the optimized CFZ-SD was evaluated for reproducibility, morphology, and physical properties and subjected to in vitro and in vivo tests. **Results**: The optimal values for each X factor were identified using a response optimization tool, achieving a yield (Y1) of 62.8%, a solubility (Y2) of 9941 μg/mL, and a particle size (Y3) of 5.89 μm, all of which were within the 95% prediction interval (PI). Additionally, amorphization induced by spray drying was confirmed for the optimized CFZ-SD using scanning electron microscopy (SEM), differential scanning calorimetry (DSC), and powder X-ray diffraction (PXRD) analyses. In in vitro dissolution tests, the final dissolution rate of the CFZ-SD increased 3.58-fold at pH 1.2 and 3.84-fold at pH 6.8 compared to an Invokana^®^ tablet. In addition, relative to CFZ, it showed an 8.67-fold and 8.85-fold increase at pH 1.2 and pH 6.8, respectively. The in vivo pharmacokinetic behavior of CFZ and the CFZ-SD was evaluated in Sprague–Dawley rats following oral administration at a dose of 5 mg/kg. The AUC of the CFZ-SD increased 1.9-fold compared to that of CFZ. **Conclusions**: In this study, a solid dispersion (SD) formulation of CFZ, a BCS class IV SGLT2 inhibitor, was developed and optimized using a QbD approach to enhance solubility and oral bioavailability.

## 1. Introduction

Hundreds of millions of people worldwide are at risk of developing diabetes, with type 2 diabetes mellitus being a chronic and progressive disease [1]. Crystalline canagliflozin (CFZ) is a sodium–glucose co-transporter 2 (SGLT2) inhibitor that received approval from the United States Food and Drug Administration (FDA) in April 2013, becoming the first agent in this class to be approved for clinical use [2]. This SGLT2 inhibitor lowers blood glucose by enhancing urinary glucose excretion [3]. In addition, this glucose-lowering effect occurs independently of insulin-mediated pathways [4,5]. According to the Biopharmaceutics Classification System (BCS), CFZ falls under class IV, being characterized by poor solubility, limited permeability, and low bioavailability [3,6]. Significant first-pass metabolism in the liver additionally contributes to its low bioavailability [6]. The absolute oral bioavailability of CFZ has been reported to be approximately 65%, which plays a role in determining its clinical dosing schedule [7]. For the effective management of type 2 diabetes mellitus, CFZ is commonly administered as a once-daily oral tablet at either 100 mg or 300 mg, in combination with dietary modifications and physical activity, in accordance with current clinical guidelines [8,9].

Spray drying, first mentioned in 1860, is a technique used to convert various liquids such as solutions, suspensions, and emulsions into powders in a single step [10]. The technique involves atomizing a liquid into small droplets and then rapidly drying them with hot air [11]. This technique can enhance the solubility and oral bioavailability of poorly water-soluble crystalline drugs, particularly by preparing amorphous solid dispersions (SDs) [12]. It is also regarded as a reliable and cost-efficient process for pharmaceutical manufacturing [13]. In an amorphous SD, the polymer not only enhances the solubility of the drug but also provides stability to prevent it from recrystallizing [14]. Careful selection of polymeric carriers is therefore essential to ensure both the enhanced dissolution and long-term stability of the formulation [15].

The objective of this study was to enhance the solubility and bioavailability of CFZ using a spray drying technique with a Quality-by-Design (QbD) approach. The Box–Behnken design (BBD), a type of response surface methodology (RSM), is widely used to methodically evaluate the effects of formulation variables and their interactions on critical response parameters [16,17]. The BBD is recognized for its high efficiency in RSM, offering comprehensive experimental information while requiring fewer runs than the widely used central composite design (CCD) [18]. For this study, the formulation of CFZ-loaded solid dispersions (CFZ-SDs) was optimized using a BBD with three factors at three levels, resulting in a total of fifteen experiments, including three central point replicates. The optimized CFZ-SD was subjected to characterization, which included yield (Y1), solubility (Y2), and particle size (Y3); differential scanning calorimetry (DSC); powder X-ray diffractometry (PXRD); scanning electron microscopy (SEM); in vitro dissolution testing; and in vivo pharmacokinetic evaluation in a rat model.

## 2. Materials and Methods

### 2.1. Materials

Canagliflozin (CFZ) was supplied by Klasia Co., Ltd. (Seoul, Republic of Korea). Invokana^®^ was purchased from Janssen Pharmaceuticals Inc. (Beerse, Belgium). Tadalafil was kindly provided by Hanmi Pharmaceutical Co., Ltd. (Hwaseong, Republic of Korea). Hydroxypropyl-β-cyclodextrin (HP-β-CD) was purchased from Roquette (Lestrem, France). α-cyclodextrin (α-CD), β-cyclodextrin (β-CD), and γ-cyclodextrin (γ-CD) were supplied by Ashland Inc. (Wilmington, DE, USA). Gelatin (Type A), sodium carboxymethyl cellulose (Na-CMC), and xanthan gum were purchased from Sigma Aldrich (Saint Louis, MO, USA). Carbomer homopolymer type B (Synthalen E83P) was provided by 3V Sigma (Georgetown, SC, USA). Polyvinyl alcohol–polyethylene glycol graft copolymer (Kollicoat IR), copovidone (Kollidon VA64), and povidone (PVP K-90) were obtained from BASF (Ludwigshafen, Germany). Polyethylene glycol (PEG 4000 and PEG 6000), maltodextrin, hypromellose (HPMC P645), hydroxypropyl cellulose (HPC L-type), and sodium alginate (Duksan, EP grade) were kindly provided by Hanmi Pharmaceutical Co., Ltd. (Hwaseong, Republic of Korea). Silicon dioxide (SiO_2_) was supplied by Boryung Co., Ltd. (Seoul, Republic of Korea). Acetonitrile, acetone, and pectin (EP grade) were purchased from Daejung Co., Ltd. (Siheung, Republic of Korea). The deionized water used in the laboratory was produced by means of a distillation device. All other chemicals used were of analytical grade.

### 2.2. HPLC Condition

An HPLC analysis of CFZ in the samples was performed using an Agilent 1260 Infinity HPLC system (Agilent Technologies, Santa Clara, CA, USA) equipped with a UV–Vis detector (Agilent G1314 1260, Agilent Technologies, CA, USA). CFZ was separated through a reversed-phase column (VDSpher PUR 100 C18-M-SE, 5 μm, 4.6 mm × 150 mm, VDS optilab, Berlin, Germany). The mobile phase consisted of a mixture of 0.1% (*v*/*v*) phosphoric acid in distilled water and acetonitrile (45:55, *v*/*v*). The HPLC analysis was performed at a flow rate of 1.0 mL/min. The injection volume was 20 μL, and UV detection was monitored at 290 nm [19]. The HPLC conditions for the CFZ plasma concentration analysis were adapted from the previous method with minor modifications. A mobile phase of 0.1% (*v*/*v*) phosphoric acid in distilled water and acetonitrile (60:40, *v*/*v*) was used, with an injection volume of 50 μL. Data acquisition and processing were performed using OpenLab CDS Chemstation LC software (product version: 2.18.18).

### 2.3. Drug Solubility Test

The solubility of CFZ was determined in distilled water (D.W.) and solutions adjusted to pH 1.2, 4.0, and 6.8. The pH 1.2 solution was prepared using 0.1 M hydrochloric acid and sodium chloride, while the pH 4.0 solution was prepared using a 0.05 M sodium acetate buffer. The pH 6.8 solution was prepared by mixing 0.2 M potassium dihydrogen phosphate with 0.2 M sodium hydroxide solution. To determine the saturation solubility of the drug, 10 mg of CFZ was added to 1 mL of each solution. The mixtures were then shaken in a shaking water bath at 37 °C and 50 rpm for 5 days. After centrifugation at 13,500 rpm for 10 min, the supernatant was filtered through a 0.45 μm syringe filter to remove insoluble CFZ [20]. All samples were diluted with 50% acetonitrile before being quantified using an HPLC system.

### 2.4. Polymer Screening

The polymers used for CFZ-SD preparation were chosen from those commonly applied in SDs to ensure suitable compatibility with the drug [21]. For polymer selection, 10 mg of CFZ was added to individual microcentrifuge tubes, each containing 1 mL of a 1% (*w*/*v*) solution of a different polymer. Notably, the polymer screening included not only high molecular weight polymers but also functional oligosaccharide such as cyclodextrins. The subsequent experimental procedures were conducted according to the method outlined in Section 3.3.

### 2.5. Identification of CQAs, CMAs, and CPPs

Among the various quality attributes, three factors were selected as CQAs, namely, yield (Y1), solubility (Y2), and particle size (Y3), because they have a significant influence on overall product quality. Based on their potential influence on these CQAs, the SiO_2_ ratio (*w*/*w*), HP-β-CD ratio (mol/mol), and blower setting were identified as CPPs and CMAs. These CPPs and CMAs were designed as X factors, while the CQAs served as Y factors. The levels of all factors were determined in preliminary tests (Table 1).

### 2.6. Production of CFZ Solid Dispersion via Spray Drying

CFZ-SDs were prepared using the spray drying technique with HP-β-CD and SiO_2_. SDs can be prepared using various methods, with the spray drying process being rapid, continuous, and able to maintain reproducibility during scale-up, making it widely used in the industrial preparation of SDs [22]. The CFZ-SDs were prepared using a spray dryer (Yamato ADL311SA; Yamato Scientific Co., Ltd., Tokyo, Japan). Spray drying was performed with an inlet temperature of 75 °C and an outlet temperature of 50 °C. The feed solution was introduced at a flow rate of 1.2 mL/min, and atomization was performed under an air pressure of 0.1 MPa. CFZ was dissolved in acetone, and HP-β-CD was dissolved in water. The resulting solutions were combined, after which SiO_2_ was dispersed into the mixture. The final mixture was then subjected to spray drying. The acetone-to-water ratio in the final solution was set to 7:3 (*v*/*v*), as determined in the preliminary experiments.

### 2.7. Yield

Each sample’s yield percentage following spray drying was assessed by comparing the weight of the resulting powder to the combined weight of the components used before the process:$$Yield\;(\%)=\frac{Final\;weight\;(mg)}{Initial\;weight\;(mg)}\times 100$$

### 2.8. Solubility

For solubility studies, CFZ-SDs equivalent to 10 mg of CFZ were placed in a microcentrifuge tube containing 1 mL of pH 1.2 solution, given that CFZ demonstrated pH-independent solubility. The mixture was then saturated by storing it in a shaking water bath (Dae Han Lab Tech, LSB-045S, Daehan Labtech, Namyangju, Republic of Korea) at 50 rpm and 37 °C for 5 days. The subsequent experimental procedures followed the same methods described in Section 3.3. The maximum plasma concentration (T_max_) of CFZ was reached at 1 h, and, considering that gastric pH is approximately 1.2, the solubility of CFZ was measured at pH 1.2 to reflect gastric conditions [23].

### 2.9. Particle Size

The particle size of each CFZ-SD was measured three times using a laser diffraction particle size analyzer (Mastersizer 3000; Malvern, Worcestershire, UK). An appropriate amount of each sample was loaded into an Aero-S tray for analysis. The measurements were conducted under the following conditions: the hopper gap was set to 1 mm, the air pressure was maintained at 1.0 bar, and the sample feed rate was kept at 75% [24,25]. Data were collected using Mastersizer 3000 v.3.00 software. Dv(50) represents the median particle size, where 50% of the total particle volume is composed of particles smaller than this value and 50% is composed of particles larger than this value. The average of the Dv(50) values obtained from three measurements was used for an ANOVA.

### 2.10. Morphological and Physicochemical Characterization

#### 2.10.1. Scanning Electron Microscopy (SEM)

The morphological features and surface structures of the samples were observed using a Tescan-MIRA3 scanning electron microscope (TESCAN KOREA, Seoul, Republic of Korea). The specimens were affixed to the sample holder with double-sided adhesive tape. To ensure electrical conductivity, a platinum layer was deposited using an EmiTech K575X Sputter Coater (EmiTech, Madrid, Spain) at a deposition rate of 6 nm/min under a vacuum of 7 × 10^−3^ mbar [26].

#### 2.10.2. Differential Scanning Calorimetry (DSC)

The thermal properties of the CFZ, CFZ-SD, and PM of the excipients were examined using DSC (Q200, TA Instruments, New Castle, DE, USA). Each sample (5 mg) was accurately weighed into an aluminum pan sealed with an aluminum lid (TA Instruments, USA) using an electric weighing balance. Sealing was performed by using a special pan-and-lid locking system. The prepared sample was placed in the DSC at its designated location, and an empty pan was used as a reference. The DSC analyses were performed over a temperature range of 30–300 °C, with the temperature increasing at a rate of 10 °C per minute. The nitrogen gas flow rate was maintained at 40 mL/min [27].

#### 2.10.3. Powder X-Ray Diffractometer (PXRD)

A PXRD analysis was performed to evaluate the crystallinity of CFZ and the CFZ-SD using a powder X-ray diffractometer (D/MAX-2500; Rigaku, Tokyo, Japan). The measurements were carried out with Cu-Ka radiation (I = 1.54178 Å) at an operating voltage of 40 kV and a current of 40 mA. Diffraction patterns were recorded over a 2θ range of 2° to 60°, with a scanning rate of 0.02° per second [28].

### 2.11. Dissolution Test

Dissolution tests of CFZ, the Invokana^®^ tablet, and the CFZ-SD were conducted using a USP apparatus II (RCZ-6N; Pharmao Industries Co., Liaoyang, China). The dissolution of the CFZ-SD was conducted only for the optimal composition. The dissolution medium was maintained at 37 ± 0.5 °C, with the paddle rotation speed set to 100 rpm throughout the experiment. CFZ, the Invokana^®^ tablet, and the CFZ-SD were exposed to 900 mL of dissolution media adjusted to pH 1.2 and pH 6.8. Aliquots were collected from the dissolution medium at predetermined time points (5, 10, 15, 20, 30, 45, 60, 90, and 120 min) over a total duration of 2 h [29]. Each collected sample was filtered using a 0.45 μm syringe filter and diluted with 50% acetonitrile. The CFZ concentration in the filtered samples was quantified using the HPLC conditions outlined in Section 3.2.

### 2.12. In Vivo Pharmacokinetic Study

Male Sprague–Dawley rats (8–9 weeks old, 250 ± 20 g) were purchased from Samtako Co. (Osan, Republic of Korea) for an in vivo pharmacokinetic study of the CFZ-SD. Prior to the experiment, the rats were acclimated to standard laboratory conditions (25 ± 2 °C, 12/12 h light/dark cycle) for one week with ad libitum access to food and water. The animal study protocol was approved by the Institutional Animal Care and Use Committee (IACUC) of Gyeongsang National University (Approval No. GNU-250409-R0077) in compliance with NIH guidelines and the Animal Welfare Act. The Sprague–Dawley rats were randomly allocated into two groups, each comprising four rats. CFZ and the CFZ-SD were suspended in 1 mL of 0.5% (*w*/*v*) CMC-Na and administered orally at a dose of 5 mg/kg. Blood samples of 350 µL were collected via the jugular vein at predetermined time points of 0.5, 1, 2, 3, 6, 9, 12, 24, 36, and 48 h after dosing, followed by immediate centrifugation at 13,500 rpm for 15 min at 4 °C. The separated plasma was harvested and stored at −20 °C until subsequent quantitative analysis. To analyze the plasma samples, 50 µL of internal standard solution (tadalafil 50 µg/mL in acetonitrile) and 300 µL of acetonitrile were added to 150 µL of plasma. The samples were subjected to vortex mixing for 3 min in order to facilitate deproteinization and drug extraction. Subsequently, they were centrifuged at 13,500 rpm for 15 min at 4 °C. The obtained supernatant was filtered through a 0.2 μm syringe filter and transferred to analytical vials for an HPLC analysis. The quantification of CFZ in plasma was performed using the second set of HPLC conditions described in Section 3.2. The following pharmacokinetic parameters were determined through a non-compartmental analysis and the area under the plasma concentration–time curve from 0 to 48 h (AUC_0–48_): maximum plasma concentration (C_max_), time to reach C_max_ (T_max_), elimination half-life (T_1/2_), and elimination rate constant (K_el_).

## 3. Results and Discussion

### 3.1. Solubility of CFZ

The solubility of a drug is a major factor in its absorption and bioavailability [30]. The saturation solubility of CFZ was measured under pH 1.2, 4.0, and 6.8 conditions representing physiological environments [31]. The saturation solubility of CFZ was found to be very low, approximately 10 μg/mL, regardless of pH (Figure 1).

### 3.2. Polymer Screening

For polymer selection, 10 mg of CFZ was added to various microcentrifuge tubes, each containing 1 mL of a different 1% (*w*/*v*) polymer solution. A total of 18 different polymers were tested in this manner. Among these, HP-β-CD, β-CD, and γ-CD solutions showed a higher solubility for CFZ than the others. This enhancement is attributed to the formation of inclusion complexes with the drug, which enhances its aqueous solubility and chemical stability [32]. Among the tested solutions, the HP-β-CD solution exhibited the highest saturation solubility of CFZ, reaching 1524.19 ± 3.83 μg/mL (Figure 2). This enhancement is likely due to the distinctive molecular structure of HP-β-CD, which features a hydrophobic inner cavity and a hydrophilic outer surface. Such a configuration enables the inclusion of poorly water-soluble drugs, thereby enhancing their aqueous solubility [33].

### 3.3. Selection of Factors and Levels in Box–Behnken Design

A BBD allows for an investigation of not only the individual influence of each factor, but also the combined effects arising from factor interactions, with non-linear relationships captured by quadratic terms [34,35]. A Minitab19^®^ (Minitab Inc., State College, PA, USA) was utilized for the optimization of the CFZ-SD using a BBD. High (+1), medium (0), and low (−1) levels were assigned to each X factor: the SiO_2_ ratio, HP-β-CD ratio, and blower. The level of SiO_2_ was determined to optimize yield during spray drying. SiO_2_ has a porous structure, and its high surface area and surface -OH groups allow it to interact with drugs [36]. Due to these structural characteristics, it acts as an excipient to alleviate the problem of drugs sticking to the walls or agglomerating during the spray drying process, thereby improving yield [37]. If the proportion of SiO_2_ is too low, then wall deposition will occur during drying, resulting in poor recovery. Conversely, if the proportion is too high, then the viscosity of the suspension will increase, making spray drying unfeasible [38]. In addition, the HP-β-CD ratio was carefully selected to balance the solubilization capacity with patient-centric formulation requirements. An excessive ratio of HP-β-CD would necessitate larger tablet dimensions due to an increased powder mass, which could potentially reduce swallowing ease and medication adherence [39]. A total of 15 experimental runs were generated using the Minitab19^®^ based on a BBD, comprising 12 factorial points and 3 center point replicates (Table 2). The replication of center points allows for the estimation of experimental error and enhances the reliability of experimental data [40].

### 3.4. Spray Drying Process Applied with Box–Behnken Design

#### 3.4.1. Yield

The regression model for yield (Y1) showed an R^2^ value of 97.15% based on an analysis of variance (ANOVA), demonstrating that the model fit the experimental data very well. Furthermore, the model was statistically significant according to the ANOVA (*p* < 0.05). Through a Pareto chart, among the main effects, the blower was identified as the most significant factor influencing the yield, followed by the quadratic term of the blower (Figure 3C). However, no other interaction or quadratic terms showed statistical significance (*p* > 0.05). This indicates that, aside from the quadratic term of the blower, the remaining interaction and quadratic terms did not have a meaningful impact on the yield. Additionally, a lack-of-fit test for yield produced a *p*-value above 0.05, indicating that the model fit the data well (Table 3). The following presents the model equation using coded terms:Y1 (Yield) = − 2.422 + 0.275X_1_ + 0.117X_2_ + 0.953X_3_ + 0.134X_1_^2^ − 0.0147X_2_^2^ − 0.0677X_3_^2^ + 0.0717X_1_X_2_ − 0.1183X_1_X_3_ − 0.0258X_2_X_3_

The yield was determined as the percentage of solids recovered relative to the amount of solids initially introduced during the spray drying process. During the spray drying process, as the blower setting was increased from 4 to 6, the yield significantly improved from 26.2% to 66.6%. The contour plot shown in Figure 3A displays distinct color changes, and the surface plot shown in Figure 3B reveals steep slope variations as the blower setting changes, visually demonstrating the high sensitivity of the yield to the blower parameter.

#### 3.4.2. Solubility

The regression model for solubility (Y2) showed an R^2^ value of 99.84% based on an ANOVA, demonstrating that the model fit the experimental data very well. The ANOVA revealed that the fitted response surface model was highly significant (*p* < 0.05). A response surface analysis revealed that HP-β-CD (*p* < 0.05) and its quadratic term (*p* < 0.05) were the most influential factors affecting solubility (Table 4 and Figure 4C). The solubility values observed in the experiments ranged from 2547 μg/mL to 9963 μg/mL, indicating a wide variation depending on the HP-β-CD ratio. The solubility increased proportionally with the HP-β-CD concentration up to a certain ratio, after which the increase slowed down, indicating a saturation effect. This trend is consistent with the quadratic effect observed in the response surface analysis. Furthermore, a lack-of-fit test for solubility produced a *p*-value above 0.05, indicating that the model fit the data well. The contour plot shown in Figure 4A displays distinct color changes, and the surface plot shown in Figure 4B reveals steep slope variations as the HP-β-CD setting changes. This observation is consistent with the response surface analysis, which indicates that solubility is most affected by the HP-β-CD parameter.

These findings have significant implications for pharmaceutical development, as improved solubility can enhance the bioavailability and therapeutic efficacy of the drug [30]. An increased solubility allows for better absorption in the gastrointestinal tract, potentially leading to reduced dosage requirements and minimized side effects. Therefore, the observed increase in solubility with HP-β-CD is expected to contribute significantly to the overall effectiveness and clinical potential of the drug formulation [41]. The following presents the model equation using coded terms:Y2 (solubility) = − 6319 + 303X_1_ + 13,227X_2_ + 1135X_3_ − 1960X_1_^2^ − 3421X_2_^2^ − 146X_3_^2^ − 246X_1_X_2_ + 520X_1_X_3_ + 139X_2_X_3_

#### 3.4.3. Particle Size

The regression model for particle size (Y3) showed an R^2^ value of 90.81% based on an ANOVA, demonstrating that the model fit the experimental data very well. Additionally, the ANOVA revealed that the fitted response surface model was highly significant. The particle sizes (Dv(50)) in the 15 formulations spanned from 4.92 μm (F9) to 8.86 μm (F4). Among the independent variables, the HP-β-CD ratio had the most significant impact on particle size (*p* < 0.05), followed by the SiO_2_ ratio, which also showed a considerable influence (*p* < 0.05). In addition, the quadratic terms of both HP-β-CD and SiO_2_ were statistically significant (Table 5). As the ratio of HP-β-CD and the blower increased, the particle size of the CFZ-SDs tended to increase [42]. As shown in Figure 5A,B, the particle size of the CFZ-SDs tended to increase with increasing ratios of HP-β-CD, SiO_2_, and the blower. Furthermore, a lack-of-fit test for particle size produced a *p*-value above 0.05, indicating that the model fit the data well. The following presents the model equation using coded terms:Y3 (particle size) = 1.54 − 5.64X_1_ + 1.44X_2_ + 1.180X_3_ + 6.28X_1_^2^ + 0.491X_2_^2^ − 0.440X_2_X_3_

### 3.5. Optimization of CFZ-SD Using BBD

The response optimization of CFZ-SD was performed based on the results of 15 formulations, focusing on CQAs such as yield (Y1), solubility (Y2), and particle size (Y3). The optimized conditions identified through the response optimization were SiO_2_ at 0.38 (*w*/*w*), HP-β-CD at 1.63 (mol/mol), and a blower setting of 5.839. However, due to practical limitations in achieving these exact values, the conditions in actual experiments were rounded to SiO_2_ at 0.4 (*w*/*w*), HP-β-CD at 1.6 (mol/mol), and a blower setting of 6. Under these adjusted conditions, the yield (Y1), solubility (Y2), and particle size (Y3) were measured at 62.8%, 9941 μg/mL, and 5.89 μm, respectively. These experimental values were all within the 95% PI (yield: 54.87–74.53%; solubility: 9313–10,630 μg/mL; particle size: 5.657–7.665 μm) predicted by the model and showed similar results to the predicted values (yield: 64.7%; solubility: 9971 μg/mL; particle size: 6.661 μm) (Table 6). These results confirm the reproducibility of the process and validate the reliability and predictive power of the optimization model, demonstrating the effectiveness of the BBD. A design space is a multidimensional combination and a range of key variables that affect quality [43]. As shown in Figure 6, the design space was obtained through the BBD, and the white area represents the process conditions that meet the optimal product characteristics [44].

### 3.6. Morphological and Physiochemical Characterization of CFZ-SD

The morphological and physical characteristics of the CFZ-SD were evaluated. Figure 7 shows SEM micrographs of CFZ, HP-β-CD, the CFZ-SD, and the physical mixture (PM). CFZ exhibited an irregular crystalline morphology, whereas the CFZ-SD displayed a smooth, spherical shape, with an average diameter of approximately 6 μm. The PM was prepared based on the optimal formulation obtained through the BBD, and CFZ was observed on the surface of HP-β-CD without any significant morphological changes.

The thermal properties of CFZ, HP-β-CD, the PM, and the CFZ-SD were analyzed using DSC (Figure 8A). The measurements were conducted over a temperature range of 30 °C to 300 °C. CFZ and the PM exhibited a distinct melting peak at 88 °C, corresponding to its crystalline melting point. In contrast, no CFZ-associated melting peak was observed in the CFZ-SD, indicating successful amorphization of the drug within the SD system [24].

The PXRD patterns of the pure CFZ, the PM of CFZ and excipients, and the CFZ-SD were compared (Figure 8B). Multiple sharp diffraction peaks indicative of crystallinity were observed for the pure CFZ, and a similar pattern was observed for the PM, suggesting that CFZ remained crystalline even after mixing. However, these crystalline peaks disappeared in the PXRD pattern of the CFZ-SD, confirming that it had transitioned to an amorphous state. Therefore, the spray drying process converted the crystalline structure of CFZ into an amorphous form [45,46]. This conversion occurs by dissolving the drug in a solvent, disrupting the crystal lattice structure, and then rapidly evaporating the solvent [47].

### 3.7. In Vitro Dissolution Test of CFZ-SD

An in vitro dissolution study of the CFZ-SD was conducted for 2 h in pH 1.2 and pH 6.8 buffer solutions, simulating physiological gastrointestinal conditions. To assess the dissolution behavior of the CFZ-SD, comparative analyses were performed under the same conditions using CFZ and a commercial formulation (Invokana^®^ tablet). The dissolution of the CFZ-SD reached 85% within 15 min at both pH 1.2 and pH 6.8, whereas neither CFZ nor the Invokana^®^ tablet reached this level (Figure 9) [48]. Because the dissolution of CFZ can be considered pH-independent (Figure 1), there was little difference in the dissolution rates of the CFZ-SD at pH 1.2 and 6.8 [28]. The final dissolution rate of the CFZ-SD increased by 3.58-fold at pH 1.2 and by 3.84-fold at pH 6.8 compared to the Invokana^®^ tablet (*p* < 0.05). In addition, relative to CFZ, it showed an 8.67-fold and 8.85-fold increase at pH 1.2 and pH 6.8, respectively (*p* < 0.05). This improvement is interpreted as a result of enhanced solubility due to both the amorphization induced by spray drying and the interaction between HP-β-CD and the drug [49,50]. These findings are consistent with the fact that the CFZ-SD formulation exhibited a markedly superior dissolution profile to both CFZ and the commercial product. Therefore, this formulation is expected to improve the poor solubility of CFZ, potentially enhancing oral absorption.

### 3.8. In Vivo Pharmacokinetic Study

Figure 10 and Table 7 present the mean plasma concentration–time profiles and pharmacokinetic parameters (AUC_0–48_, C_max_, T_max_, T_1/2_, and K_el_) following the oral administration of CFZ and the CFZ-SD to Sprague Dawley rats at an equivalent dose of 5 mg/kg. CFZ had an AUC_0–48_ of 7675.32 ± 595.33 ng·h/mL and a C_max_ of 870.05 ± 33.95 ng/mL. The CFZ-SD demonstrated an AUC_0–48_ of 14,650.43 ± 2383.81 ng·h/mL, which was 1.9-fold higher than that observed for CFZ (*p* < 0.05). The CFZ-SD also showed a C_max_ of 1089.59 ± 199.22 ng/mL, which was 1.1-fold higher than that observed for CFZ (Table 7). In addition, the CFZ-SD exhibited a longer T_max_ than CFZ, which is attributed to the presence of high concentrations of CD. As CD is not absorbed through the gastrointestinal tract and only the free drug dissociated from the complex can be absorbed, a high level of CD can lower the free drug concentration and consequently reduce the absorption rate [51,52]. However, in the optimized CFZ-SD, although T_max_ was slightly extended, the AUC increased. These results suggest that the bioavailability increased as the solubility of the CFZ-SD manufactured through spray drying increased, as shown in the in vitro test results. CFZ and the CFZ-SD showed no significant difference in blood concentration up to 3 h; however, after 6 h, the CFZ-SD showed a significantly higher blood concentration than CFZ (Figure 10). The results of the in vivo pharmacokinetic study demonstrate that spray drying using the BBD successfully improved the bioavailability of CFZ, a poorly soluble drug.

This study confirmed that the bioavailability of CFZ was improved through enhanced solubility. While the improved solubility of CFZ through CD and spray drying was demonstrated in vitro, the effect of CD on permeability was not established. In in vivo evaluations, the CFZ-SD exhibited approximately a 1.9-fold increase in AUC compared to CFZ. However, the contribution of permeability enhancement could not be confirmed, and the improvement in AUC was attributed to increased solubility [53,54]. If the permeability of CFZ-SD did not increase, a greater increase in AUC could be expected with the enhancement of permeability in future studies. Therefore, further studies are warranted to verify whether the permeability of the CFZ-SD is indeed altered and to elucidate its impact on bioavailability enhancement.

## 4. Conclusions

In this study, a solid dispersion formulation of CFZ, a BCS class IV SGLT2 inhibitor, was developed and optimized using a QbD approach to enhance solubility and oral bioavailability. A BBD was employed to optimize critical quality attributes such as yield (Y1), solubility (Y2), and particle size (Y3) and to determine the relationships between factors (X) and these responses (Y). The formulation demonstrated successful amorphization of the drug, as confirmed via SEM, DSC, and PXRD, and it exhibited significantly improved in vitro dissolution rates. Additionally, in vivo results showed a significant improvement in the AUC, indicating that this formulation also improved bioavailability. The results demonstrate that QbD-based optimization effectively produced SD formulations with improved quality attributes, thereby improving the solubility of poorly water-soluble drugs and thus increasing bioavailability.

## Figures and Tables

**Figure 1 pharmaceutics-17-01319-f001:**
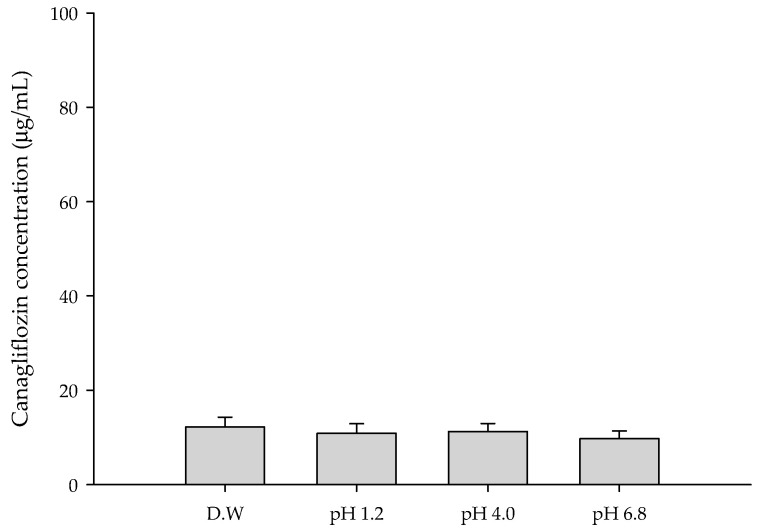
Saturation solubility of CFZ in various aqueous solutions.

**Figure 2 pharmaceutics-17-01319-f002:**
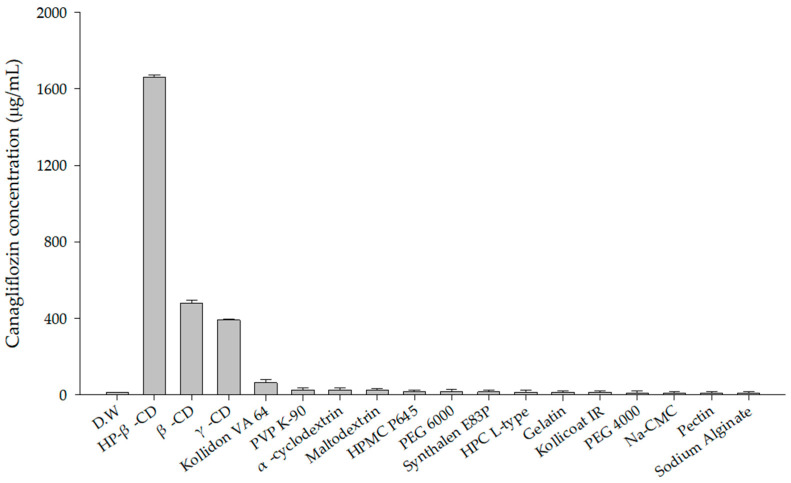
Solubility of CFZ in various 1% (*w*/*v*) polymer solutions.

**Figure 3 pharmaceutics-17-01319-f003:**
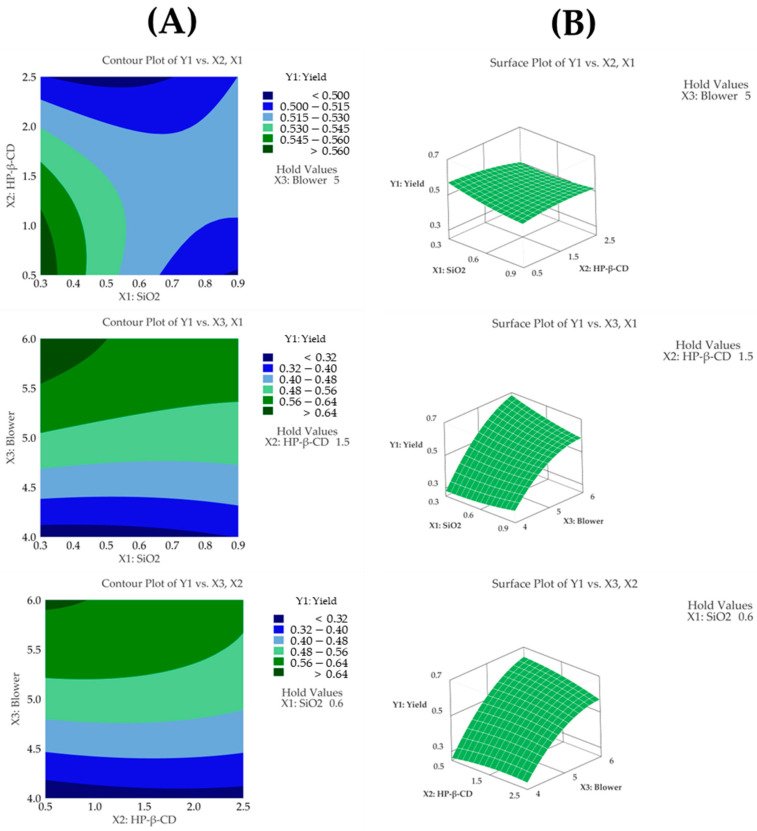
(**A**) Contour plot, (**B**) surface plot, and (**C**) Pareto chart for Y1 yield.

**Figure 4 pharmaceutics-17-01319-f004:**
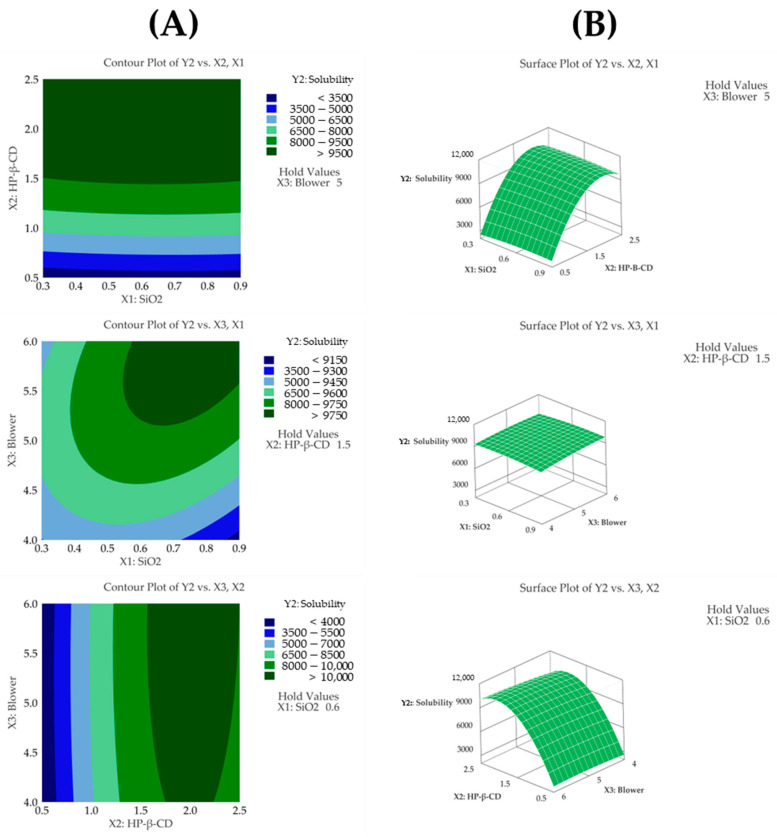
(**A**) Contour plot, (**B**) surface plot, and (**C**) Pareto chart for Y2 solubility.

**Figure 5 pharmaceutics-17-01319-f005:**
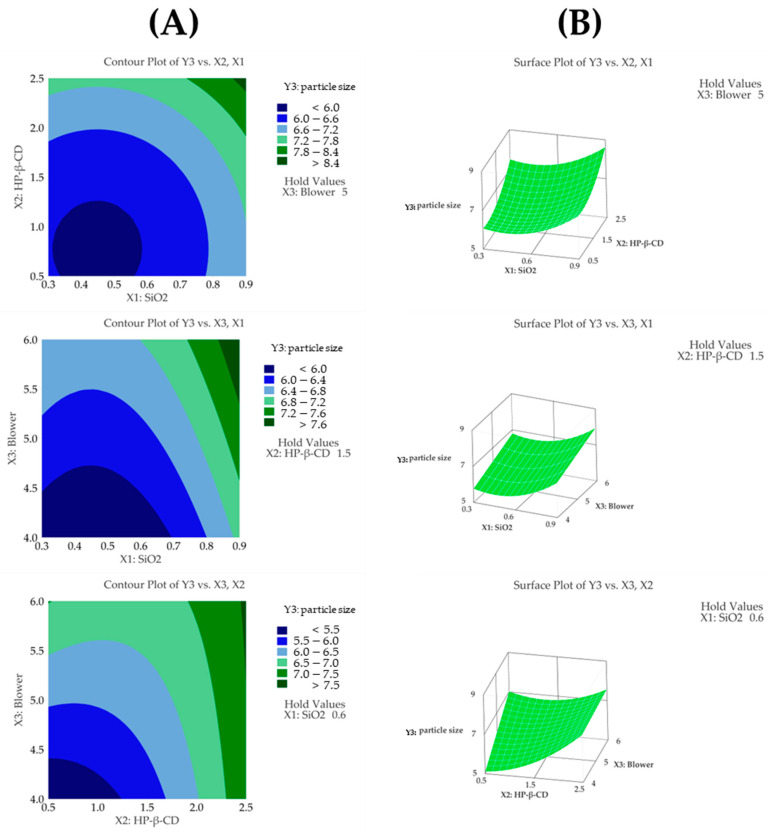
(**A**) Contour plot, (**B**) surface plot, and (**C**) Pareto chart for Y3 particle size.

**Figure 6 pharmaceutics-17-01319-f006:**
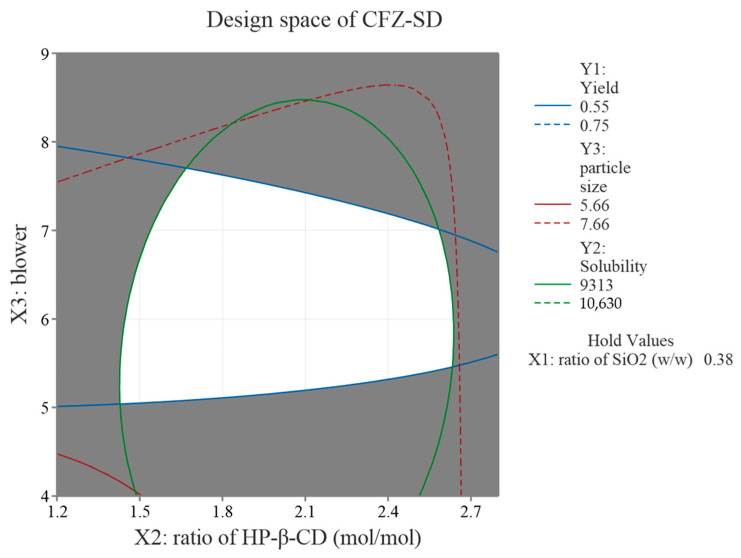
Overlaid contour plot of CFZ-SD.

**Figure 7 pharmaceutics-17-01319-f007:**
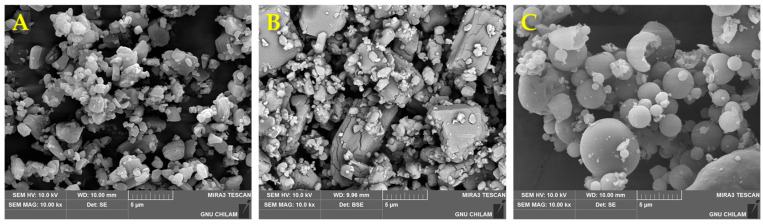
Scanning electron micrographs: (**A**) CFZ (×10,000), (**B**) PM (×10,000), and (**C**) CFZ-SD (×10,000).

**Figure 8 pharmaceutics-17-01319-f008:**
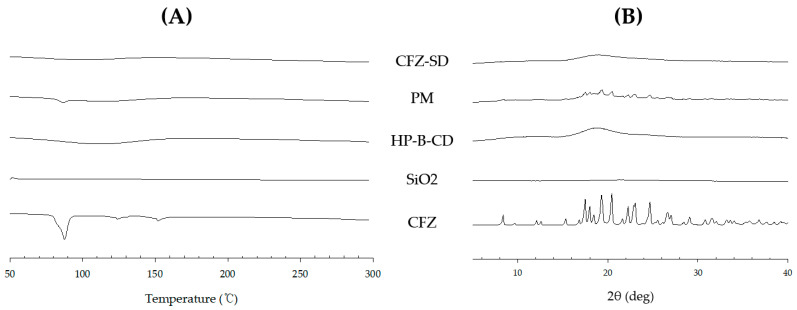
(**A**) DSC thermograms and (**B**) PXRD patterns of CFZ-SD, PM, HP-β-CD, SiO_2_, and CFZ.

**Figure 9 pharmaceutics-17-01319-f009:**
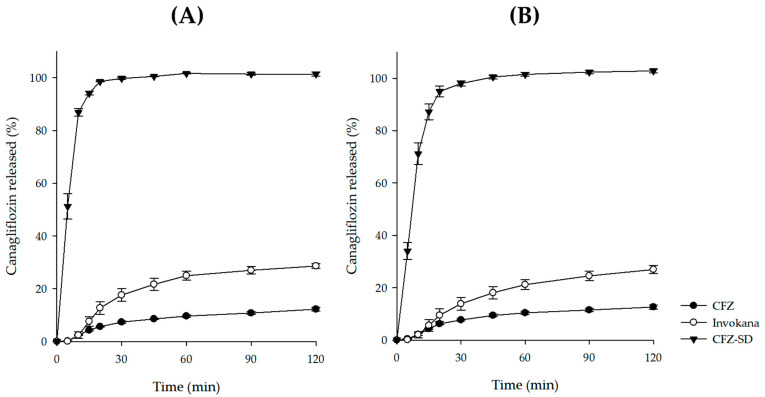
Dissolution profile of CFZ, Invokana, and CFZ-SD at (**A**) pH 1.2 and (**B**) pH 6.8.

**Figure 10 pharmaceutics-17-01319-f010:**
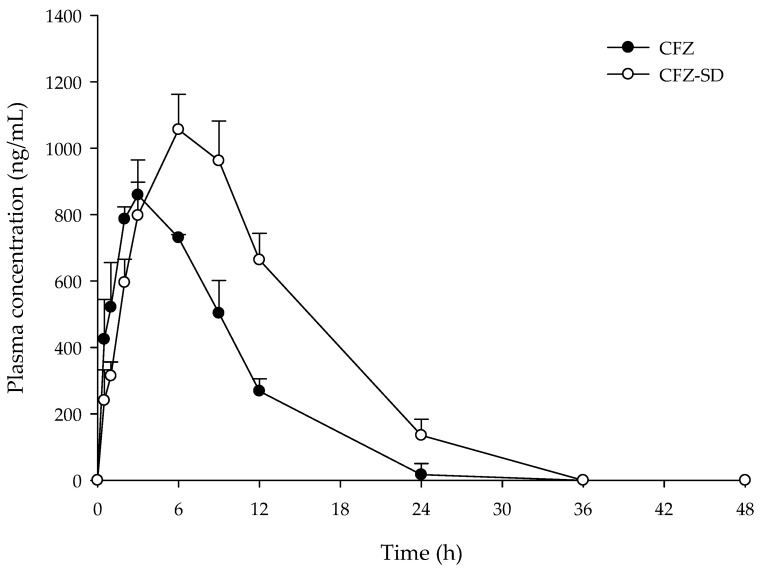
Plasma concentration–time profiles of CFZ and CFZ-SD in rats.

**Table 1 pharmaceutics-17-01319-t001:** Factor (X) levels and response (Y) goals in the Box–Behnken design.

X Factors	Variation Intervals	
	Low	High
X_1_: SiO_2_ ratio (*w*/*w*)	1:0.3	1:0.9
X_2_: polymer ratio (mol/mol)	1:0.5	1:2.5
X_3_: blower	4.0	6.0
Y factors	Goal	
Y1: yield (%)	Maximize	
Y2: solubility (μg/mL)	Maximize	
Y3: particle size (μm)	Minimize	

**Table 2 pharmaceutics-17-01319-t002:** Factor (X) levels and response (Y) results in the Box–Behnken design.

Run	X Factors			Y Factors		
Unit	X_1_: SiO_2_ Ratio (*w*/*w*)	X_2_: HP-β-CD Ratio (mol/mol)	X_3_: Blower	Y1: Yield (%)	Y2: Solubility (μg/mL)	Y3: Particle Size (μm)
F1	0.3	0.5	5	58.5	2547	5.67
F2	0.9	0.5	5	46.3	2577	7.61
F3	0.3	2.5	5	53.6	9806	7.39
F4	0.9	2.5	5	50.0	9541	8.86
F5	0.3	1.5	4	26.5	9135	6.06
F6	0.9	1.5	4	35.7	9160	6.87
F7	0.3	1.5	6	65.0	9314	6.98
F8	0.9	1.5	6	60.0	9963	7.33
F9	0.6	0.5	4	26.2	2667	4.92
F10	0.6	2.5	4	26.8	9325	7.16
F11	0.6	0.5	6	66.6	2694	7.19
F12	0.6	2.5	6	56.9	9908	7.67
F13	0.6	1.5	5	54.8	9743	6.65
F14	0.6	1.5	5	51.3	9481	6.16
F15	0.6	1.5	5	51.0	9921	6.21

**Table 3 pharmaceutics-17-01319-t003:** ANOVA outcomes and model summary for Y1 yield.

Source	DF *	Adj SS *	Adj MS *	F-Value	*p*-Value
Model	9	0.2531	0.0281	18.97	0.002
Linear Model	3	0.2251	0.0750	50.61	0.000
X_1_: SiO_2_ ratio (*w*/*w*)	1	0.0016	0.0016	1.13	0.336
X_2_: HP-β-CD ratio (mol/mol)	1	0.0013	0.0013	0.89	0.388
X_3_: blower	1	0.2221	0.2221	149.79	0.000
Quadratic Model	3	0.0184	0.0061	4.15	0.080
X_1_: SiO_2_ ratio	1	0.0005	0.0005	0.36	0.574
X_2_: HP-β-CD ratio	1	0.0007	0.0007	0.54	0.496
X_3_: blower	1	0.0169	0.0169	11.42	0.020
Two-Way Interaction	3	0.0095	0.0031	2.15	0.213
X_1_: SiO_2_ ratio X_2_: HP-β-CD ratio	1	0.0018	0.0018	1.25	0.315
X_1_: SiO_2_ ratio X_3_: blower	1	0.0050	0.0050	3.40	0.125
X_2_: HP-β-CD ratio X_3_: blower	1	0.0026	0.0026	1.79	0.239
Error	5	0.0074	0.0015		
Lack-of-fit	3	0.0065	0.0022	4.87	0.175
Pure error	2	0.0009	0.0004		
S		R		R^2^ (Retouch)	
0.0385069		97.15%		92.03%	

* DF: degrees of freedom; Adj SS: adjusted sum of squares; Adj MS: adjusted mean squares.

**Table 4 pharmaceutics-17-01319-t004:** ANOVA outcomes and model summary for Y2 solubility.

Source	DF	Adj SS	Adj MS	F-Value	*p*-Value
Model	9	142,486,449	15,831,828	338.35	0.000
Linear Model	3	99,007,026	33,002,342	705.30	0.000
X_1_: SiO_2_ ratio (*w*/*w*)	1	24,090	24,090	0.51	0.505
X_2_: HP-β-CD ratio (mol/mol)	1	98,666,128	98,666,128	2108.62	0.000
X_3_: blower	1	316,808	316,808	6.77	0.048
Quadratic Model	3	43,283,039	14,427,680	308.34	0.000
X_1_: SiO_2_ ratio	1	114,861	114,861	2.45	0.178
X_2_: HP-β-CD ratio	1	43,208,809	43,208,809	923.43	0.000
X_3_: blower	1	78,301	78,301	1.67	0.252
Two-Way Interaction	3	196,384	65,461	1.40	0.346
X_1_: SiO_2_ X_2_: HP-β-CD ratio	1	21,756	21,756	0.46	0.526
X_1_: SiO_2_ ratio X_3_: blower	1	97,344	97,344	2.08	0.209
X_2_: HP-β-CD ratio X_3_: blower	1	77,284	77,284	1.65	0.255
Error	5	233,959	46,792		
Lack-of-fit	3	135,983	45,328	0.93	0.557
Pure error	2	97,976	48,988		
S		R		R^2^ (Retouch)	
216.314		99.84%		99.54%	

**Table 5 pharmaceutics-17-01319-t005:** ANOVA outcomes and model summary for Y3 particle size.

Source	DF	Adj SS	Adj MS	F-Value	*p*-Value
Model	6	11.5399	1.9233	13.17	0.001
Linear Model	3	8.8208	2.9403	20.13	0.000
X_1_: SiO_2_ ratio (*w*/*w*)	1	2.6106	2.6106	17.88	0.003
X_2_: HP-β-CD ratio (mol/mol)	1	4.0470	4.0470	27.71	0.001
X_3_: blower	1	2.1632	2.1632	14.81	0.005
Quadratic Model	2	1.9447	0.9724	6.66	0.020
X_1_: SiO_2_ ratio	1	1.1881	1.1881	8.14	0.021
X_2_: HP-β-CD ratio	1	0.8939	0.8939	6.12	0.038
Two-Way Interaction	1	0.7744	0.7744	5.30	0.050
X_2_: HP-β-CD ratio X_3_: blower	1	0.7744	0.7744	5.30	0.050
Error	8	1.1682	0.1460		
Lack-of-fit	6	1.0228	0.1705	2.34	0.329
Pure error	2	0.1454	0.0727		
S		R		R^2^ (Retouch)	
0.382138		90.81%		83.91%	

**Table 6 pharmaceutics-17-01319-t006:** Optimized spray drying settings and predicted values.

Factor			Setting		
X_1_: SiO_2_ ratio (*w*/*w*)			1:0.4		
X_2_: HP-β-CD ratio (mol/mol)			1:1.6		
X_3_: blower			6		
**Response**	**Suitable Value SE**	**Suitable Value**		**95% CI**	**95% PI**
Y1: yield	0.6470	0.0219		(0.5983, 0.6958)	(0.5487, 0.7453)
Y2: solubility	9971	137		(9619, 10,324)	(9313, 10,630)
Y3: particle size	6.661	0.209		(6.179, 7.143)	(5.657, 7.665)

**Table 7 pharmaceutics-17-01319-t007:** PK parameters.

PK Parameter	CFZ-SD	CFZ
AUC_0–48_ (ng·h/mL)	14,650.43 ± 2383.81	7675.32 ± 595.33
C_max_ (ng/mL)	1089.59 ± 199.22	870.05 ± 33.95
T_max_ (h)	5.97 ± 0.88	3.11 ± 0.64
T_1/2_	4.69 ± 1.01	4.48 ± 1.10
K_el_	0.16 ± 0.03	0.17 ± 0.05

## Data Availability

Data are available on request due to restrictions, e.g., privacy or ethical restrictions.

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
