# Peer review of "Research on Enhancing the Solubility and Bioavailability of Canagliflozin Using Spray Drying Techniques with a Quality-by-Design Approach"

_pharmaceutics, 2025, doi:10.3390/pharmaceutics17101319_

Round 1
Reviewer 1 Report
Comments and Suggestions for Authors
This work summarizes the principle of spray drying technology and the potential to improve the solubility and bioavailability of poorly water-soluble crystalline drugs. By detailing the optimization of CFZ-loaded solid dispersions (CFZ-SDs) through a Box-Behnken design (BBD) with three factors at three levels, the authors explain the development of a novel formulation. The paper highlights the advantages of this approach, demonstrating its benefits through morphological, physicochemical, in vitro dissolution, and in vivo pharmacokinetic studies. These studies indicate that the optimized CFZ-SD significantly enhances solubility and oral bioavailability by inducing amorphization, offering a promising strategy for enhancing the therapeutic efficacy of poorly water-soluble drugs like CFZ. This manuscript has certain research value and can be considered for publication after minor revisions. The following issues can be optimized and modified:
- The introduction mentions that CFZ is a BCS Class IV drug but does not provide specific data or literature references for its low solubility and low permeability.
- In drug solubility testing, the selection of pH 1.2, 4.0, and 6.8 is based on simulating physiological environments. Could the authors please specify which specific parts of the human gastrointestinal tract these pH values correspond to?
- The SEM images show that the CFZ-SD forms smooth spherical particles, a morphology that contrasts sharply with the irregular crystals of the unprocessed drug. The authors should discuss how this significant change in particle morphology occurred in their study.
- The DSC and PXRD results confirm the amorphous state of the drug in the CFZ-SD. However, the physical stability of this formulation has not been evaluated. Given the potential risk of recrystallization in amorphous solid dispersions, it is essential to supplement the study with data on crystalline changes after accelerated stability testing (e.g., storage under 40°C/75% RH conditions for 1-3 months).
- The article mentions that the acetone-water ratio of the feed solution in the spray drying process is 7:3. Does maintaining this ratio ensure that CFZ-SD powders of the same quality can be obtained regardless of the processing volume, without any impact on particle morphology and drug dispersion state?
Reviewer 2 Report
Comments and Suggestions for Authors
The manuscript covers a relevant subject, although the novelty of the approach is limited. Screening for excipients that increase bioavailability of a BCS class IV drug after amorphization via spray drying without a detailed hypothesis on the selection of the excipients can be widely found in literature.
Specific comments:
The presentation of fig1 should allow a better comparison (range of the y-axis), if necessary at all.
To my opinion, cyclodextrins do not fall under the term “polymers”
The choice of the potentially solubility enhancing excipients are not coherent to my opinion and should be revised. Eudragit RS is for example not soluble under the selected conditions, gelatin is of an unclear grade, the same is true for alginates, pectinate, HPC and so on.
Fig3: should be redrawn, since most of the digits are difficult to read, the same applies to fig4
Fig7+8: morphological analyses lacks information on the characterization of the non-best formulations. While the authors still provide data on size and solubility, similar in-depth data should be created or presented of XRPD and dissolution measurements.
Fig9: PK studies should comprise the commercial product for comparison, in order to better understand whether differences between the 3 formulations in-vitro also translate into in-vivo behaviour. Beyond this, it would interesting to what extend the beforementioned variables impact oral bioavailability.
Finally, the discussion remains very descriptive, since a general discussion of the findings in the context of existing studies is not provided. Also, conclusions whether authors speculate on a higher impact of their formulation on a solubility or a permeability increase are not given. The authors only discuss solubility, however permeability modifying properties of CDs need to be addressed here, too.
Reviewer 3 Report
Comments and Suggestions for Authors
Thank you for this opportunity to review. This article titled “Research on enhancing the solubility and bioavailability of 2 Canagliflozin using spray drying techniques with a Quality- 3 by-Design approach" could contribute significant updates on a novel spray dried formulation of canagliflozin. The overall structure of the article is well-defined, however, there are some comments that I would like the authors to address before considering its acceptance. The comments are included below:
- It would be beneficial in providing a brief discussion on the rationale of including SiO2 in the spray dried formulation.
- Please elaborate on the purpose of the blower parameter for the spray drying process.
- CFZ is a BCS class 4 compound, meaning it also has low permeability. How does the currented optimized spray dried dispersion improve permeability, and what is the mechanism behind it?
- I would suggest the authors to perform statistical analysis for the dissolution profiles and PK parameters to show that optimized formulation and control are statistically significant.
- Make sure that the conclusions drawn are fully supported by the data. For instance, if claiming a “significant improvement” in bioavailability, ensure the term “significant” is backed by statistical analysis, or rephrase.
- The manuscript can benefit from stating the limitations of the scope of the study and potential future directions.
minor editing for language (e.g. ensuring consistent terminology for the formulation and eliminating any duplicate text or typos).
Round 2
Reviewer 3 Report
Comments and Suggestions for Authors
The authors have adequately answer all of the comments and made the appropriate changes to the manuscript. At the current state, I believe the manuscript is in good condition for acceptance. Thank you!